# A Cross-Sectional Questionnaire-Based Survey on Blood-Borne Infection Control among Romanian Dental Students

**DOI:** 10.3390/medicina58091268

**Published:** 2022-09-13

**Authors:** Catalina Iulia Saveanu, Gianina Dărăbăneanu, Livia Ionela Bobu, Daniela Anistoroaei, Irina Bamboi, Irina Ciortescu, Alexandra Ecaterina Saveanu

**Affiliations:** 1Department of Surgicals, Faculty of Dental Medicine, University of Medicine and Pharmacy Grigore T Popa, 700115 Iasi, Romania; 2Faculty of Dental Medicine, University of Medicine and Pharmacy Grigore T Popa, 700115 Iasi, Romania; 3Faculty of Medicine, University of Medicine and Pharmacy Grigore T Popa, 700115 Iasi, Romania

**Keywords:** infection control, blood-borne transmission, HIV, HBV, HCV

## Abstract

*Background and Objectives:* According to the CDC estimates, 5.6 million healthcare workers worldwide are exposed to the risk of occupationally contracting blood-borne pathogens, including human immunodeficiency virus (HIV), hepatitis B virus (HBV), hepatitis C virus (HCV), and others. The aim of the present study was to assess the knowledge level on blood-borne infection control (IC) among Romanian undergraduate dental students. *Materials and Methods*: A cross-sectional, questionnaire-based survey with 21 items was conducted from May 2022 to June 2022. The study had *α* = 0.620 and Cronbach’s Alpha = 0.660. The Chi-square test was used for data comparison. The data were analyzed using IBM SPSS version 26 (IBM, Armonk, NY, USA), and *p* ≤ 0.05 was considered statistically significant. *Results*: The study sample included 207 subjects with a mean age of 21.38 (±1.9) years, 59.9% F (female), 40.1% M (male), 38.2% students from year II and 61.8% from year III. Most of the subjects did not have knowledge on the indirect mode of IC, the persistence of HBV, HCV, or HIV, or the existence of rapid tests (*p* < 0.05). PEP (post-exposure prophylaxis) was known as follows: HBV, 32.36% (67); HCV, 25.60% (53); and HIV, 36.71% (76); *p* < 0.05. Only 50.24% (104) had knowledge on the recommended testing moments (*p* = 0.019 by gender, *p* = 0.752 by year of study). The optimal time to access PEP was considered by 28.01% (58), *p* ˃ 0.05. Only 37.68% (78) strongly agreed that the one-hand covering technique of the needle was efficient in IC (*p* < 0.05). *Conclusions*: The evaluation of the knowledge regarding IC highlighted major gaps in the perception of the subjects, which underlined the need to implement sustained forms of continuing medical education on this topic.

## 1. Introduction

According to the estimates of the Centers for Disease Control and Prevention (CDC), 5.6 million people working in healthcare and related fields are occupationally exposed to the risk of contracting blood-borne pathogens, such as human immunodeficiency virus (HIV), hepatitis B virus (HBV), hepatitis C virus (HCV), and others [1]. Numerous studies are described in the specialized literature about blood-borne pathogens [1,2,3,4,5,6,7,8,9,10,11]. The major risks of blood-borne infections in the dental office are percutaneous injuries caused by needles or other sharp objects, and the risk of transmission depends on the type and size of the lesion, the duration of exposure, and the concentration of the virus [12].

Studies show that up to one-third of all sharp object injuries occur during elimination. The CDC estimates that 62 to 88% of sharp object injuries can be prevented simply by using safer medical devices [1].

Hepatitis B vaccines became available in 1981 and have remained the primary method of hepatitis B prevention [13,14]. However, the risk for this disease increases from 6 to 30%, depending on its concentration at the source [1].

The risk of transmission by exposure to HCV-positive blood is 1.8%, much lower than in the case of HBV. At present, there is no vaccine or specific post-exposure prophylaxis (PEP) available for HCV infection [13], but the use of direct-acting antivirals (DAAs) to treat chronic hepatitis C has resulted in a significant increase in rates of sustained viral response (approximately 90–95%) [14].

The risk of percutaneous transmission of HIV is the lowest, when compared to HBV and HCV, estimated to be approximately 0.3% [1]. When exposure has occurred, post-exposure prophylactic drugs are very effective in preventing HIV, provided they are administered within the first 72 h [13]. Numerous studies have highlighted the importance of HIV infection control [15,16,17,18,19].

The awareness of the dental staff and their training for knowledge on the protocol corresponding to infection control and the handling of sharp instruments is particularly important; at the same time, dental staff should be trained in the protocol they should follow in the event of an injury with a sharp instrument or syringe needle, aspects highlighted in other specialized studies [1,16,17,18,19].

By strictly adhering to OSHA regulations and guidelines, it is possible to reduce the risk of exposure and prevent the transmission of blood-borne agents [12]. To this end, health officials need to develop and implement a plan of measures to reduce occupational exposure to these pathogens [13].

Post-exposure prophylaxis (PEP) comprises a set of measures applied in order to minimize the risk of developing an infectious disease among healthcare professionals as a result of potential exposure to blood-borne pathogens, and should be started as soon as possible [1]. In the event of a percutaneous injury with a needle contaminated with the blood of a HIV-positive patient, initiation of antiretroviral therapy within 1–2 h may reduce the risk of HIV transmission by 81%; in the case of exposure to HBsAg-positive blood, can reduce the risk by almost 75%; and for hepatitis C, although a specific PEP is not available, early treatment can prevent chronic infection [19].

This assessment, conforming to the CDC, should be performed in several stages, namely: immediately after exposure, then one week, three months, six months, and twelve months after exposure, depending on the type and source of exposure, and the post-exposure protocol plan should be attached to the medical record [1]. Identifying the dangers and risks to patient safety should represent a basic goal [20,21,22,23].

In this context, the aim of the present study was to assess the knowledge level on blood-borne infection control among Romanian undergraduate dental students.

## 2. Materials and Methods

The evaluation of the level of knowledge was performed by the questionnaire method.

### 2.1. Setting and Participants

A cross-sectional study was conducted from May to June 2022. The subjects were students of the Faculty of Dentistry within the University of Medicine and Pharmacy “Grigore T. Popa” in Iasi (UMFI). A total of 349 students were registered in the dentistry program conducing to a diploma of Bachelor of Dentistry in Romanian, French and English language in the II and III years of study [24].

According to the calculation formula [25], a minimum number of 184 subjects was representative of the university. A total number of 207 students were included in the study sample.

The study had statistical power alpha, α = 0.620, and Cronbach’s Alpha = 0.660.

### 2.2. The Survey

A questionnaire with 21 multiple choice questions was used for this study. The questions focused on the level of knowledge of students in the field of infection control. The questionnaire was reviewed for validity by three experts, on a number of 20 students. For this purpose, the students completed the questionnaire in its entirety and then were interviewed by three members (C.I.S.; L.B and I.B.) who corrected the interpretable aspects.

### 2.3. Study Group

This study included dental students randomly selected from the UMFI. Ethical acceptance for this study was given in No. 177/17.04.2022. The inclusion criteria were: second- and third-year students in dentistry, and subjects who had agreed to participate. The exclusion criteria was: subjects who did not agree to participate. The questionnaire was completed by 207 students.

### 2.4. Demographic Characteristics

Demographic data included: age, gender, and year of study. To complete these data, three questions were answered.

### 2.5. Domain: Knowledge Data

The 18 multiple choice questions (3, 4, or 5 options) about infection control were distributed as follows: 5 questions about the modes of infection transmission in the dental office; 5 questions for the assessment of knowledge on the persistence of the virus on surfaces; 3 questions for the assessment of knowledge about the existence of the HBV vaccine and rapid tests for HBV, HCV, and HIV infections, and 7 questions for the assessment of knowledge on the accidental post-exposure protocol in the case of infected patients (HBV, HCV, and HIV). Five-point Likert scale response options were used to evaluate some of the questions: Likert scale—level of agreement was used for questions Q4–Q8 and Likert scale—quality for questions was used for questions Q16–Q18 and Q20.

### 2.6. Data Collection

A descriptive statistic was performed for this study. Pearson’s Chi-square test was used for data comparison. Symmetric measurements: nominal by nominal (phi, Cramer’s V, contingency coefficient), interval by interval (Pearson’s r), and ordinal by ordinal (Spearman correlation) were used. These data were analyzed using IBM SPSS version 26 (IBM, Armonk, NY, USA), with a *p* ≤ 0.05.

## 3. Results

### 3.1. Demographic Data

The study included 207 subjects with a mean age of 21.38 (±1.9) (min: max = 18:31) years. The distribution of students by gender was female: male = 59.9% (124):40.1% (83), and by year of study II: III = 38.2% (79):61.8% (128).

### 3.2. Assessing the Perception of the Level of Knowledge on the Modes of Infection Transmission

The evaluation of the knowledge on the modes of infection transmission showed that a percentage of 52.65% (109) of the subjects knew very well that blood-borne infections could be transmitted by direct contact (*p* ˃ 0.05) (Table 1). A percentage of 13.52% (28) of the investigated students strongly agreed and 32.85% (68) agreed with the fact that blood-borne pathogens could be transmitted by indirect contact. No significant differences were found by gender (*p* = 0.336) or by study year (*p* = 0.463). Only 73.42% (152) of the subjects knew the obligation to wear protective equipment. Most subjects, 92.27% (191), knew that the doctor must be notified when the patient suffers from an infectious disease. Only 63.28% (131) of the subjects treated all the patients as potentially infected.

### 3.3. Assessing the Perception of the Knowledge Level about the Persistence of HBV, HCV, and HIV on Surfaces

The evaluation of the knowledge regarding the persistence of HBV on surfaces showed that most of the subjects, 83.10% (172), did not know that it could persist for more than 7 days, with no significant differences by study year (*p* = 0.083) (Table 2).

Only approximately half of the subjects, 51.20% (106), knew the persistence time of HCV on surfaces, with significant differences by gender (*p* = 0.039), but not by study year (*p* = 0.423).

Regarding the persistence of HIV, its persistence on surfaces was not very well known by 34.29% (71) of the subjects. Differences were statistically significant by gender (*p* = 0.023) in favor of males, but not by year of study (*p* = 0.071).

### 3.4. Assessing the Perception of the Knowledge Level about the Existence of the HBV Vaccine and Rapid Tests for HBV, HCV, and HIV Infections

Evaluation of the knowledge regarding the existence of the HBV vaccine showed that the subjects, in a percentage of 18.35% (38), declared that they had no knowledge on this aspect, and 9.66% (20) of them did not know the fact that a HBV vaccine exists (Table 3). The differences were significant by study year (*p* = 0.035) in favor of the third-year subjects, but not by gender (*p* = 0.475). Moreover, only 64.25% (133) of the subjects knew that rapid detection tests were available for HBV. The differences were not significant by gender (*p* = 0.072) or by study year (*p* = 0.749). Only half of the students, 49.27% (102), were aware of the existence of rapid tests for HCV detection, and 60.38% (125) were aware of the existence of rapid tests for HIV detection. No significant differences were found in this respect.

### 3.5. Assessing the Perception of the Knowledge Level Regarding the Accidental Post-Exposure Protocol for HBV-, HCV-, and HIV-Infected Blood

Evaluation of the subjects’ perceptions of knowledge regarding the accidental post-exposure protocol for patients infected with HBV (F:M/II:III), HCV (F:M/II:III), and HIV (F:M/II:III) showed that the following number of students considered that they knew very well the protocol: for HBV, 67 (33:34/49); HCV, 53 (26:27/45:18); and HIV, 76 (36:40/53:23). The following number of students stated that they were well acquainted with this protocol: for HBV, 50 (29:21/12:38); HCV, 50 (31:19/17:33); and HIV, 52 (35:17/12:40). A large part of the subjects declared that they did not know well this protocol, being undecided, as follows: for HBV, 45 (29:16/11:34); HCV, 46 (31:15/11:35); and HIV, 41 (23:18/11:30). Many of the subjects did not consider that they knew the post-exposure protocol in the case of existence of a contact with a HBV-infected person. Thus, the following number of students disagreed: for HBV, 28 (21:7/2:26); HCV, 29 (21:8/2:27); and HIV, 25 (20:5/1:24), and strongly disagreed: for HBV, 17 (12:5/5:12); HCV, 19 (15:4/4:15); and HIV, 13 (10:3/2:11). The differences in responses regarding the knowledge on the post-exposure protocol in cases of a contact with a HBV-infected person were statistically significant by study year (*p* = 0.000), but not by gender (*p* = 0.131). For the knowledge on PEP for HCV- and HIV-infected blood, the differences were significant both by gender and study year (Table 4).

### 3.6. Assessing the Perception of the Knowledge Level Regarding the Post-Exposure Attitude in the Case of a Patient Contaminated with HBV, HCV, or HIV and the Knowledge on the Technique of One-Hand Needle Cover

Only 104 (72:32/37:67) subjects knew that in the case of accidental exposure to contaminated fluid, respectively, blood, they must perform immediate testing. The others answered that the test must be performed 3 months and 6 months after the exposure if the patient had no symptoms, 27 (12:15/9:18), or 3 months, 6 months, and 9 months after the exposure if the patient had no symptoms, 41 (19:22/19:22). A number of twenty-nine subjects (19:10/12:17) did not know that a test needs to be performed, and six subjects (2:4/2:4) declared that they did not get tested. The differences were significant by gender (*p* = 0.019), but not by study year (*p* = 0.752). The optimal time to access the epidemiological service after accidental exposure to blood-contaminated objects was considered by 58 subjects (36:22/28:30) to be at least one day; 77 subjects (45:32/22:55), at most one day; 43 subjects (23:20/17:26), not more than one hour; 19 subjects (15:4/9:10), not more than one week, and 10 subjects (5:5/3:7), not more than 3 months, on the occasion of performing the analyzes. The differences were not significant either by gender (*p* = 0.380) or by year of study (*p* = 0.164). Answers to the question “Do you consider that one-hand covering technique of the needle is efficient in preventing blood-borne infections?” showed that seventy-eight (36:42/40:38) subjects strongly agreed; sixty-three (46:17/29:34) agreed; thirty-seven (25:12/7:30) were undecided; twenty-one (12:9/2:19) disagreed; and eight (5:3/1:7) strongly disagreed with this technique. The differences were significant both by gender (*p* = 0.020) and study year (*p* = 0.000) (Table 5).

## 4. Discussion

### 4.1. Assessing the Perception of the Level of Knowledge on How to Transmit Infections

Dental healthcare workers (DHCWs) have an increased risk of exposure to blood-borne pathogens such as hepatitis B virus, hepatitis C virus and HIV mostly from percutaneous injury during their daily activity. Such lesions may be produced during invasive procedures by accidental cuts which break the integrity of the skin, in which needlestick injury is the most common route for blood-borne pathogens. The fact that blood-borne infections could potentially be transmitted by indirect contact was known by only 13.52% (28) of the subjects, and is to a very large extent an aspect that should not be overlooked. As most carriers of infectious diseases cannot be identified clinically, implementation of standard universal precautions in dental schools is the most effective way to control cross-infection [2,26]. Students, such as other healthcare workers who come in contact with patients’ blood and body fluids, may be exposed to severe infections during their clinical activity [27].

In our study, only 73.42% (152) of the subjects knew the obligation of the protective equipment and only 63.28% (131) of the subjects treated all the patients as potentially infected. The CDC recommends measures included in the concept of universal precautions in order to reduce occupational exposures. According to this concept, each patient should be considered as potentially infected, as it is not possible to determine the infectivity status of each patient before treatment. These measures must also be applied by dentists.

The measures comprised by the concept of standard precautions, including the use of barrier techniques, have proven to be the best prevention strategies against the occupational transmission of infectious diseases in healthcare services. Al-Maweri et al., in Saudi Arabia, found that 98.8% and 90.8% of the dental students declared they always wore gloves and masks, respectively, during clinical activity. The use of eye protection was reported by only 29.2%. They also noticed a higher level of knowledge and better compliance with infection control measures among the students of final study years [2]. Fatima et al., in a study on dental students and interns, found that 23% stated they always complied to universal precautions while treating patients [20]. Hbibi et al., in Morocco, found that most of the dental students declared that they always wore gloves for consultations (98.8%) as well as protective masks (80.7%), while the use of protective eyewear was occasional among 77.1% of the respondents. Almost 23.2% of subjects reported knowledge on antiretroviral chemoprophylaxis [3]. A study conducted in Lithuania by Akgun showed that 56.4% of the dental students reported a moderate level of knowledge about “The Universal Precaution Guidelines” [4]. A very good level of knowledge was reported by significantly more of the younger students than students of the final years (31.3% vs. 10.0%). A percentage of 80.3% of the dental students were aware that HBV, HCV, and HIV/AIDS could be transmitted after injury with sharp instruments.

### 4.2. Assessing the Perception of the Level of Knowledge about the Persistence of HBV, HCV, and HIV on Surfaces

Hepatitis B is one of the most infectious diseases, with a transmission pathway similar to other infectious diseases. However, the present study showed that only 16.90% of the students knew the virus persistence on surfaces. This is important in terms of infection control. The results of a study conducted in Malaysia showed that 72.5% of the dental students had correct knowledge on the transmission of HBV and 54% of the interns had knowledge on the post-exposure protocol [15]. A study conducted by Khandelwal et al. among dental students in India showed that 86% of them had knowledge about hepatitis B infection. Most of the students had good knowledge regarding modes of transmission, but 21% of them did not know that saliva was a vehicle for hepatitis B transmission [11]. A study conducted by Kumar among dental students and interns in India found that 81.9% were aware that an active vaccine to prevent hepatitis B transmission was available. Only 50% of the students showed a positive opinion towards recording data on hepatitis B in the patient’s file [28]. The study conducted by Li in China pointed out that 89.43% of the dental undergraduates had good or excellent knowledge on HBV transmission, illustrating the good results of education on infection control before clinical practice [16].

The perception of the level of knowledge regarding the persistence of HCV was better, 51.20% (106), among the subjects in the present study. Regarding the persistence of HIV on surfaces, only 34.29% (71) of the students had correct knowledge. A study among dental students in Iraq showed that they had moderate knowledge on HIV, and 78% of them knew about the rapid test [16]. Another study conducted in Egypt among dental students in 3rd, 4th, and 5th years of study showed that 33.1% of them knew that HIV could survive up to 1 h outside human tissues, and 84.3% considered that infection control measures to prevent hepatitis C infection could also prevent the transmission of HIV in the dental setting. Significant differences by year of study were found concerning the opinion upon the obligation of HIV-infected patients to reveal their HIV status to their dentists [18]. The study conducted by Jin et al. in China among dental students showed that 60% of them had adequate knowledge on the prevention of HIV transmission in the dental office [19].

### 4.3. Assessing the Perception of the Level of Knowledge about the Existence of the HBV Vaccine

Dentists may be exposed to pathogens during their clinical activity, even if they apply basic preventive measures. To avoid cross-infection, some safety precautions should be taken in addition to vaccination and proper adherence to the post-exposure protocol [29].

In our study, 87.29% of the students knew about the existence of the HBV vaccine. Complete hepatitis B vaccination is the best procedure to prevent the transmission of the infection during dental treatments. The study conducted by Maltezou in Greece showed that 45.9% of the dental students presented full vaccination rates against hepatitis B [30].

### 4.4. Evaluation of the Perception of the Level of Knowledge Regarding the Accidental Post-Exposure Protocol for HBV-, HCV-, HIV-Infected Patients and the Knowledge on the Technique of One-Hand Needle Cover

El-Saaidi, in a study in four public dental schools in Egypt, among 3rd-, 4th- and 5th- year students, found that a high percentage of them did not have very good knowledge on post-exposure preventive measures. At the same time, the fourth-year students had higher scores of infection control knowledge, attitudes, and practices than the fifth-year students. This trend suggests that after becoming acquainted with clinical practice, senior students may relax their attention to infection control, or lose some of the notions acquired during the previous study years. This means that students should be re-trained in infection control before graduation [31].

Fatima, in a study among dental students and interns in India, found that 55.4% were aware that PEP is most effective if administered within 1 h of the percutaneous injury. A total of 44.5% of the students correctly answered that patient’s blood should be evaluated for HIV, HBV, and HCV infections in case the healthcare worker suffers a percutaneous injury. Concerning the duration of PEP for HIV, 32.5% of the students provided correct responses, i.e., PEP should be continued for 4 weeks [20].

A longitudinal study conducted by Huynh between 2009 and 2019 on dental students in Australia showed that half of the respondents (51.9%) reported that they were not aware of PEP, and most respondents (68.1%) also were not aware of the PEP 72-h window. There were 188 (55.3%) respondents in the present study who believed needles should be re-capped. The one-handed re-capping technique was favored by 152 (44.1%) respondents, the two-handed technique by 14 (4.1%) respondents, and the needle block or other safety devices by 52 (15.1%) respondents [22].

Ramich, in a study in Germany [15], found that 50% of the students, 13% of the dentists, and 45% of the assistants did not have proper knowledge on the standardized procedure of PEP for HIV.

PEP should preferably be administered within the first hour to be most effective. A maximum delay time of 24–72 h may be allowed; after this time, its effectiveness in infection prevention decreases considerably. In the case of HIV, this is due to the fact that the virus does not immediately infect the dendritic cells in the mucosa and skin at the site of the lesion, but appears gradually within the first 24 h. Administration of PEP within the first hour after exposure importantly limits the proliferation of the virus in dendritic cells or lymph nodes, thus, preventing systemic infection [20].

After use, the syringe needles should never be re-capped or otherwise handled with both hands and should not be directed to another part of the body or to another person. When it is necessary to re-cap the needle, this should be performed either using the one-handed technique or using a special mechanical device designed to support the needle cover so as to facilitate a one-handed re-cap. Protective devices against injuries caused by sharp objects are also indicated (e.g., needles with wrapping mechanisms) [26]. In addition, the re-capping of the needle has been banned by the National Institute for Occupational Safety and Health (NIOSH, CDC) since the 1990s [32].

Previous studies have shown that female students and female health workers are more compliant than males with infection control protocols [31]. A recent study from Croatia found that males had higher chances of underestimating such injuries [33]. In the present study, differences were found between the male and female subjects. Moreover, differences were found between students attending different years of study.

Kumar (India, 2015) [28] found significant differences between groups (interns, final-year students, and third-year students) in what concerns knowledge and preventive practices, with higher scores for the third-year students, followed by final-year students and then interns. This suggests that students may forget some of the notions learned about infection control; consequently, it is necessary to re-educate them in this regard at a time closer to graduation. In actuality, the topic of infection control requires an active approach throughout the licensing course.

The higher the level of knowledge on the pathogenesis of microorganisms and the ways to prevent accidents at work, the more positive should be the attitudes and practices in this regard, leading, implicitly, to the reduction of the frequency of accidents. In other words, knowledge is an important tool to promote adherence to protocols [34,35].

Educational institutions play a key role in student’s attitudes about the adoption of correct habits for the control of cross-infection. Therefore, there is a great need for more incisive biosafety awareness by students to acquire the perception of risk and build the capacity of protection, making them understand the need for care and caution in performing dental procedures, shaping them into practicing safe clinical day to day behavior.

Prevention is the main and most effective measure to avoid the occupational transmission of diseases in dental practice. Considering that, the preventive practices of dentistry students need to be improved.

Future dental practitioners can avoid such accidents if they reinforce the standard precautions against all patients, maintain vigilance regarding contamination sources, replace some high-risk habits (e.g., re-capping needles) with more secure practices [36], and engage in a regular continuing education program throughout their careers.

More research is needed to identify the possibilities to enhance the safety conditions in dental practice, to improve students’ awareness regarding infection control, and to modulate educational methods related to the current concepts of prevention. The current educational technology, such as virtual reality, could help the dental student to improve his/her dexterity to perform appropriate tasks in a safe environment [37].

Surgical procedures could even be learned with secure tools and give more confidence to the student in high-risk situations. Using such technologies for infection control education may have a large perspective since the primary feedback seems to be satisfactory [32].

### 4.5. Study Limitations

The clinical relevance of this study consisted in the fact that the lack of knowledge on the assessed aspects entailed the exposure of the members of the medial team to a high biological risk, with a lack of control in the transmission of infection.

The limitations of this study mattered in the fact that this was a cross-sectional questionnaire study.

Another limitation of this study was given by the small number of participants, the uneven distribution by gender, our study having more female subjects, the random selection of subjects and year of study, and the lack of assessment of bias.

## 5. Conclusions

Assessment of the knowledge on how to transmit infections showed that only half of the subjects knew that blood-borne pathogens could be transmitted only by direct contact, and did not know the persistence times of blood-borne pathogens on surfaces, PEP, or the existence of rapid detection tests for HBV, HCV, and HIV.

Less than a quarter of the subjects did not know the obligation of protective equipment, and were unaware of the availability of the vaccine and the optimal time to access the epidemiological service after accidental exposure to blood-contaminated objects.

Following this study, we could affirm the fact that more comprehensive and better individualized educational measures are needed to provide working conditions with minimal risk for future professionals.

## Figures and Tables

**Table 1 medicina-58-01268-t001:** Assessing the perception of the knowledge level on the modes of infection transmission. Comparative data and Chi-square correlations.

	Gender				Study Year			
	F	M	*p*	*r*	Second	Third	*p*	*r*
	N (%)	N (%)			N (%)	N (%)		
Q4 = Do you consider that blood-borne infections can be transmitted through direct contact?								
Strongly agree	67 (32.36)	42 (20.28)	0.475	3.521	40 (19.32)	69 (33.33)	0.785	1.734
Agree	46 (22.22)	29 (14.00)			29 (14.00)	46 (22.22)		
Undecided	7 (3.38)	9 (4.34)			8 (3.86)	8 (3.86)		
Disagree	4 (1.93)	2 (0.96)			2 (0.96)	4 (1.93)		
Strongly disagree	0	1 (0.48)			0	1 (0.48)		
Q5 = Do you consider that blood-borne infections can be transmitted through indirect contact?								
Strongly agree	20 (9.66)	8 (3.86)	0.336	4.558	8 (3.86)	20 (9.66)	0.463	3.597
Agree	44 (21.25)	24 (11.59)			25 (12.07)	43 (20.77)		
Undecided	26 (12.56)	26 (12.56)			18 (6.66)	34 (16.42)		
Disagree	26 (12.56)	20 (9.66)			22 (10.62)	24 (11.59)		
Strongly disagree	8 (3.86)	5 (2.41)			6 (2.89)	7 (2.59)		
Q6 = Do you consider that protection equipment is mandatory if the patient has no symptoms?								
Strongly agree	89 (42.99)	63 (30.43)	0.348	4.456	70 (33.81)	82 (39.61)	0.001 *	18.137
Agree	29 (14.00)	15 (7.24)			5 (2.41)	39 (18.84)		
Undecided	5 (2.41)	3 (1.44)			3 (1.44)	5 (2.41)		
Disagree	0	2 (0.96)			1 (0.48)	1 (0.48)		
Strongly disagree	1 (0.04)	0			0	1 (0.48)		
Q7 = Do you consider that the patient is obliged to announce to the doctor if he/she is suffering from an infectious disease?								
Strongly agree	116 (56.03)	75 (36.23)	0.197	4.682	70 (33.81)	121 (58.45)	0.340	3.358
Agree	7 (3.38)	4 (1.93)			7 (3.38)	4 (1.93)		
Undecided	0	3 (1.44)			1 (0.48)	2 (0.96)		
Disagree	1 (0.48)	1 (0.48)			1 (0.48)	1 (0.48)		
Strongly disagree	0	0			0	0		
Q8 = Do you consider that, in general, all patients should be treated as infected?								
Strongly agree	81 (39.13)	50 (2.41)	0.495	3.387	47 (22.70)	84 (40.57)	0.633	2.564
Agree	28 (13.52)	23 (11.11)			20 (9.66)	31 (114.81)		
Undecided	12 (5.79)	5 (2.41)			7 (2.59)	10 (4.83)		
Disagree	2 (0.96)	4 (1.93)			4 (1.93)	2 (0.96)		
Strongly disagree	1 (0.48)	1 (0.48)			1 (0.48)	1 (0.48)		

N = count; * *p* = significance level; *r* = Pearson’s correlation.

**Table 2 medicina-58-01268-t002:** Assessing the perception of the knowledge level about the persistence of HBV, HCV, and HIV on surfaces. Comparative data and Chi-square correlations.

	Gender				Study Year			
	F	M	*p*	*r*	Second	Third	*p*	*r*
	N (%)	N (%)			N (%)	N (%)		
Q9 = What is the persistence time of HBV on a surface?								
A few minutes	11 (5.31)	12 (5.79)	0.159	5.184	6 (2.89)	17 (8.21)	0.083	6.680
A few hours	42 (20.28)	34 (16.42)			24 (11.59)	52 (25.12)		
Less than 3 days	45 (21.73)	28 (13.52)			36 (17.39)	37 (17.87)		
More than 7 days	26 (12.56)	9 (4.34)			13 (6.28)	22 (10.62)		
Q10 = What is the persistence time of HCV on a surface?								
A few seconds	3 (1.41)	9 (4.34)	0.039 *	8.376	2 (0.96)	10 (4.83)	0.423	2.804
A few minutes	20 (9.66)	18 (8.69)			16 (7.72)	22 (10.62)		
At most 10 h	67 (32.36)	39 (18.84)			40 (19.32)	66 (31.88)		
At least 16 h	34 (16.42)	17 (8.21)			21 (10.14)	30 (14.49)		
Q11 = What is the persistence time of HIV on a surface?								
A few seconds	11 (5.31)	15 (7.24)	0.023 *	9.576	10 (4.83)	16 (7.72)	0.071	7.042
Less than 30 min	20 (9.66)	15 (7.24)			20 (9.66)	15 (7.24)		
Less than 60 min	41 (19.80)	34 (16.42)			27 (13.04)	48 (23.18)		
More than 90 min	52 (25.12)	19 (9.17)			22 (10.62)	49 (23.67)		

N = count; * *p* = significance level; *r* = Pearson’s correlation.

**Table 3 medicina-58-01268-t003:** Assessing the perception of the knowledge level about the existence of the HBV vaccine and rapid tests for HBV, HCV, and HIV infections. Comparative data and Chi-square correlations.

	Gender				Study Year			
	F	M	*p*	*r*	Second	Third	*p*	*r*
	N (%)	N (%)			N (%)	N (%)		
Q12 = Does a vaccine against hepatitis B virus exist?								
Yes	87 (42.02)	62 (29.95)	0.475	1.490	65 (31.40)	84 (40.57)	0.035 *	6.727
No	11 (5.31)	9 (4.34)			5 (2.41)	15 (7.24)		
I do not know	26 (12.56)	12 (5.79)			9 (4.34)	29 (14.00)		
Q13 = Do you know that quick tests exist for detecting HBV infection?								
Yes	73 (35.26)	60 (28.98)	0.072	5.261	53 (25.60)	80 (38.64)	0.749	0.578
No	9 (4.34)	7 (3.38)			5 (2.41)	11 (5.31)		
I do not know	42 (20.28)	16 (7.72)			21 (10.14)	37 (17.87)		
Q14 = Do you know that quick tests exist for detecting HCV infection?								
Yes	63 (30.43)	39 (18.84)	0.144	3.876	32 (15.45)	70 (33.81)	0.077	5.139
No	11 (5.31)	15 (7.24)			14 (6.76)	12 (5.79)		
I do not know	50 (24.15)	29 (14.00)			33 (15.94)	46 (22.22)		
Q15 = Do you know that quick tests exist for detecting HIV infection?								
Yes	78 (37.68)	47 (22.70)	0.151	5.297	54 (26.08)	71 (34.29)	0.282	3.815
No	18 (8.69)	22 (10.62)			12 (5.79)	28 (13.52)		
I do not know	27 (13.04)	14 (6.76)			13 (6.28)	28 (13.52)		

N = count; * *p* = significance level; *r* = Pearson’s correlation.

**Table 4 medicina-58-01268-t004:** Assessing the perception of the knowledge level regarding the accidental post-exposure protocol for HBV-, HCV-, and HIV-infected blood. Comparative data and Chi-square correlations.

	Gender				Study Year			
	F	M	*p*	*r*	Second	Third	*p*	*r*
	N (%)	N (%)			N (%)	N (%)		
Q16 = Do you consider that you know the post-exposure protocol in the case of existence of contact with a HBV-infected person?								
Strongly agree	33 (15.94)	34 (16.42)	0.131	7.090	49 (23.67)	18 (8.69)	0.000 *	54.529
Agree	29 (14.00)	21 (10.14)			12 (5.79)	38 (18.35)		
Undecided	29 (14.00)	16 (7.72)			11 (5.31)	34 (16.42)		
Disagree	21 (10.14)	7 (3.38)			2 (0.96)	26 (12.56)		
Strongly disagree	12 (5.79)	5 (2.41)			5 (2.41)	12 (5.79)		
Q17 = Do you consider that you know the post-exposure protocol in the case of existence of contact with a HCV-infected person?								
Strongly agree	26 (12.56)	37 (17.87)	0.005 *	15.031	45 (21.73)	18 (8.69)	0.000 *	48.237
Agree	31 (14.97)	19 (9.17)			17 (8.21)	33 (15.94)		
Undecided	31 (14.97)	15 (7.24)			11 (5.31)	35 (16.90)		
Disagree	21 (10.14)	8 (3.86)			2 (0.96)	27 (13.04)		
Strongly disagree	15 (7.24)	4 (1.92)			4 (1.93)	15 (7.240		
Q18 = Do you consider that you know the post-exposure protocol in the case of existence of contact with a HIV-infected person?								
Strongly agree	36 (17.39)	40 (19.32)	0.016 *	12.177	53 (25.60)	23 (11.11)	0.000 *	54.574
Agree	35 (16.90)	17 (8.21)			12 (5.79)	40 (19.32)		
Undecided	23 (11.11)	18 (8.69)			11 (5.31)	30 (14.49)		
Disagree	20 (9.66)	5 (2.41)			1 (0.48)	24 (11.59)		
Strongly disagree	10 (4.83)	3 (1.44)			2 (0.96)	11 (5.31)		

N = count; ** p* = significance level; *r* = Pearson’s correlation.

**Table 5 medicina-58-01268-t005:** Assessing the perception of the knowledge level regarding the post-exposure attitude in the case of a patient contaminated with HBV, HCV, or HIV and the knowledge on the technique of one-hand needle cover. Comparative data and Chi-square correlations.

	Gender				Study Year			
	F	M	*p*	*r*	Second	Third	*p*	*r*
	N (%)	N (%)			N (%)	N (%)		
Q19 = The post-exposure protocol implies testing for HBV, HCV, and HIV at the following intervals of time.								23
Immediately, at 3 months, and at 6 months	72 (34.78)	32 (15.45)	0.019 *	11.737	37 (17.87)	67 (32.36)	0.752	1.910
At 3 and at 6 months if the patient has no symptoms	12 (5.79)	15 (7.24)			9 (4.34)	18 (8.690)		
At 3 months, at 6 months and at 9 months if the patient has no symptoms	19 (9.17)	22 (10.62)			19 (9.17)	22 (10.62)		
I do not know	19 (9.17)	10 (4.83)			12 (5.79)	17 (8.21)		
I am not testing	2 (0.96)	4 (1.93)			2 (0.96)	4 (1.93)		
Q20 = Do you consider that the one-hand covering technique of the needle is efficient in preventing blood-borne infections?								
Strongly agree	36 (17.39)	42 (20.28)	0.020 *	11.643	40 (19.32)	38 (18.35)	0.000 *	22.679
Agree	46 (22.22)	17 (8.21)			29 (14.00)	34 (16.42)		
Undecided	25 (12.07)	12 (5.79)			7 (3.38)	30 (14.49)		
Disagree	12 (5.79)	9 (4.34)			2 (0.96)	19 (9.17)		
Strongly disagree	5 (2.41)	3 (1.44)			1 (0.48)	7 (3.38)		
Q21 = What is the optimal time to access the epidemiological service after accidental exposure to blood-contaminated objects?								
At least one day	36 (17.39)	22 (10.62)	0.380	4.196	28 (13.52)	30 (14.49)	0.164	6.514
At most one day	45 (21.73)	32 (15.45)			22 (10.62)	55 (26.57)		
Not more than one hour	23 (11.11)	20 (9.66)			17 (8.21)	26 (12.56)		
Not more than one week	15 (7.24)	4 (1.93)			9 (4.34)	10 (4.83)		
Not more than 3 months when we perform analyzes	5 (2.41)	5 (2.41)			3 (1.44)	7 (3.38)		

N = count; ** p* = significance level; *r* = Pearson’s correlation.

## Data Availability

Not applicable.

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
