# Peer review of "A Cross-Sectional Questionnaire-Based Survey on Blood-Borne Infection Control among Romanian Dental Students"

_medicina, 2022, doi:10.3390/medicina58091268_

Round 1

Reviewer 1 Report

Abstract

The abstract is over the word limit and it does not read clearly. Please consolidate statistics for better understanding of the conclusions of the study.

Methods

1. It is not clear from methods how the study was designed. For example, the formula used to calculate statistical power (line 141) is missing and linked reference does not explicitly convey parameters used. Please define formulae used, including the one to calculate Cronbach’s coefficient, define parameters used including, covariance, reliability, etc., and how an alpha of 0.620/0.660 was achieved.

2. It is unclear what the role of the interviewers were. In line 150, please clarify was what “corrected the interpretable aspects” means in the context of the study.

Results

1. Conclusion from result 3.3 “most of the subjects, 83.10% (172) do not know that it can persist for more than 7 days” contradicts what was described in the introduction (line 61-63) “The virus has a stable structure, and remains viable and infective for long periods, up to 1 week on surfaces”. There is no acceptable answer provided in the questionnaire that accounts for the information which is, according to the introduction, up to 7 days.

2. Please define the “one-hand needle cover” technique (line 243)

Minor comments:

Line 39 – Centers for Disease Control and Prevention

Line 42 – Reference missing (estimates from CDC report)

Line 42 – Blood-borne

Line 42 – Please clarify “approached in numerous specialized studies”

Line 55 – Reference cited incorrectly

Line 56 – Please provide reference for this statement

Line 56 – Please provide reference for this statement “Nurses are particularly at risk”

Line 57 – Please provide reference for this statement “. The CDC estimates that 62 to 88% of sharp object injuries”

Line 60 – Please provide reference for this statement “The incidence of HBV infection is continuously increasing in the general population”

Line 61 – Please clarify what “state structure means in this context”

Line 63 – Please provide reference for this statement “up to 1 week on 62 surfaces”

Line 63 – Please provide reference for this statement “risk of HBV infection after percutaneous exposure varies from 6 to 30%”

Line 71 – This statement is not true. HIV-1 can be transmitted via blood contact only via percutaneous inoculations.

Line 73 – Please provide reference for this statistic “estimated to be about 0.3%”

Line 79 – This statement is redundant (“there are no known cases of aerosol transmission of a blood-borne pathogen in a clinical setting”)

Line 81 – Please provide reference for “in studies…”

Author Response

Dear reviewer,

Thank you very much for agreeing to review this article. We also thank you for the indications given to improve this article. We will respond briefly to the correction instructions suggested by you.

Abstract

  1. The abstract is over the word limit and it does not read clearly. Please consolidate statistics for better understanding of the conclusions of the study.

Response 1 I reduced the number of words to 269 out of 300 maxima. I have removed the informed consent number. I have modified the statistics to make them clearer. I applied total percentages to each question, leaving the number of cases in parentheses.

Methods

  1. It is not clear from methods how the study was designed. For example, the formula used to calculate statistical power (line 141) is missing and linked reference does not explicitly convey parameters used. Please define formulae used, including the one to calculate Cronbach’s coefficient, define parameters used including, covariance, reliability, etc., and how an alpha of 0.620/0.660 was achieved.

Methods Responces 1

For the selection of the study group, we took into account the recommendations from the education and health guidelines. Practically, through our study we aimed to evaluate a level of knowledge, so according to the recommendations we applied the calculation formula

The calculated sample size it was made with formula for confidence level p=95%, z=1.96, with margin of error by 5%, by population size N=349. For this formula I mentioned the source.

where

z is z score
pÌ‚ is the population proportion
n and n' are sample size
N is the population size

I calculated Cronbach's alpha this with SPSS program, as below.

The link I gave reference number 24 was not for how I calculated the Cronbach's alpha Coefficient. The link was for lot size. I calculated the lot size with the formula given in reference 24, namely:

I rephrased in text

  1. It is unclear what the role of the interviewers were. In line 150, please clarify was what “corrected the interpretable aspects” means in the context of the study.

Methods - Response 2

The interpretable aspects in the context of the study were where the subjects asked questions to better understand what they had to answer.

Results

  1. Conclusion from result 3.3 “most of the subjects, 83.10% (172) do not know that it can persist for more than 7 days” contradicts what was described in the introduction (line 61-63) “The virus has a stable structure, and remains viable and infective for long periods, up to 1 week on surfaces”. There is no acceptable answer provided in the questionnaire that accounts for the information which is, according to the introduction, up to 7 days.

Results – Response 1

The studies on the persistence of the hepatitis B virus are diverse, it is shown that it persists for up to 7 days and more than 7 days, the period being variable.

I change the phrase .

”The virus has a stable structure, and remains viable and infective for long periods, on surfaces”

  1. Please define the “one-hand needle cover” technique (line 243)

Results – Response 2

The unimanual needle re-covering technique aims to touch the syringe with a safe hand. After the anesthetic puncture, the needle is withdrawn, applied inside the cap. This is how the cover is fixed. Then the cover is fixed with the opposite hand.

https://www.registerednursern.com/one-hand-scoop-technique/

Minor comments:

Line 39 – Centers for Disease Control and Prevention  

Responces - Thank you, I have changed this.

Line 42 – Reference missing (estimates from CDC report)

Responces – Thank you, I have changed this. I changed source 31 to position 1 and I renumbered.

Line 42 – Blood-borne,

Responces - Thank you, I made the correction

Line 42 – Please clarify “approached in numerous specialized studies”

Responces – Thanks, I reworded it like this.

Numerous studies are described in the specialized literature about bloodborne pathogens.

Line 55 – Reference cited incorrectly

Responces – Thank you, I have changed this.

Line 56 – Please provide reference for this statement “Nurses are particularly at risk”

Responces - I have removed this sentence.

Line 57 – Please provide reference for this statement “. The CDC estimates that 62 to 88% of sharp object injuries”

Responces - I have completed the reference. Thanks!

Line 60 – Please provide reference for this statement “The incidence of HBV infection is continuously increasing in the general population”

Responces - Responces - I have completed the reference. Thanks!

Line 61 – Please clarify what “state structure means in this context”

Responces -  I change

The virus does not change its structure and remains viable and infectious on surfaces for long periods of time.”

Line 63 – Please provide reference for this statement “up to 1 week on 62 surfaces”

Responces -I add the reference

Line 63 – Please provide reference for this statement “risk of HBV infection after percutaneous exposure varies from 6 to 30%”

Responces --I add the reference

Line 71 – This statement is not true. HIV-1 can be transmitted via blood contact only via percutaneous inoculations.

Responces – I change

”HIV-1 can be transmitted via blood contact by via percutaneous inoculations”

Line 73 – Please provide reference for this statistic “estimated to be about 0.3%”

Responces - I add the reference

Line 79 – This statement is redundant (“there are no known cases of aerosol transmission of a blood-borne pathogen in a clinical setting”)

Responces -  I correct

”Thus, the risk of transmitting HIV infection in a dental setting is given in principal by the contact with the blood due to percutaneous lesions and, to a lesser extent, contact with the mucosa and skin”

Line 81 – Please provide reference for “in studies…”

Responces - I add the reference

We thank you for your help in improving the manuscript and if there are still aspects to be clarified, we will do so with pleasure.

Best regards,

Assoc. Professor, MD, PhD, Iulia Saveanu

Reviewer 2 Report

Line 22 – we should avoid to write about No of ethical acceptance in the abstract – it shoul be only mentioned in the main paper body in the Methods section

Line 25 – please write about only one gender e.x. female, because in this case there is more women

Line 25 – “from years II and III (38.2% and 61.8%, respectively)” – I don’t understand – what does this percent mean? You should clarify

Line 47 – “Transmission can occur”  - it’s not grammatically correct, maybe can be transmitted in this contect?

Line 69 – I agree that there is no vaccine available or post exposure drugs, but you should mention that DAA’s are very effcective, almost 100%

The abstract is too long – you should move some paragraphs to the discussion section or remove a few ones especially those with very basic medical knowledge

The questionnaire with all questions should be added in the appendix

The study limitations should be at the end of the discussion

Author Response

Dear Reviewer,

Thank you very much for the kindness of making this review during this holiday period. Your evaluations are very to the point and we thank you for the suggestions. We will respond to your requests point by point.

Question 1

Line 22 – we should avoid to write about No of ethical acceptance in the abstract – it shoul be only mentioned in the main paper body in the Methods section

Response 1

I have removed the informed consent number. Thanks!

In addition, I made the following changes for the abstract.

I have modified the statistics to make them clearer. I applied total percentages to each question, leaving the number of cases in parentheses.

Question 2

Line 25 – please write about only one gender e.x. female, because in this case there is more women

Response 2

In order not to discriminate between genders and because it was not easy to read, we applied female and male response percentages without differentiation. The detailed results will be presented in the results chapter.

Question 3

Line 25 – “from years II and III (38.2% and 61.8%, respectively)” – I don’t understand – what does this percent mean? You should clarify

Response 3

Thank you, I have changed to the following phrase.

”The study sample included 207 subjects with a mean age of 21.38 (± 1.9) years, 59.9% F (female), 40.1% M (male), 38.2% students from years II and 61.8% from year III.”

Question 4

Line 47 – “Transmission can occur”  - it’s not grammatically correct, maybe can be transmitted in this contect?

Response 4

We changed the phrase. Thank you!

” Transmission of infection can be from a patient to a dental healthcare provider (DHCP), from a DHCP to a patient, or from one patient to another (cross-infection).”

Question 5

Line 69 – I agree that there is no vaccine available or post exposure drugs, but you should mention that DAA’s are very effective, almost 100%

Response 5

Thank you very much for this addition suggestion. I added the following phrase and went to the bibliography.

” At present, there is no vaccine or specific post-exposure prophylaxis (PEP) available for HCV infection [12], but the use of direct-acting antivirals (DAAs) to treat chronic hepatitis C has resulted in a significant increase in rates of sustained viral response (around 90%-95%) [13].”

Question 6

The abstract is too long – you should move some paragraphs to the discussion section or remove a few ones especially those with very basic medical knowledge

Response 6

Thank you, I reduced the number of words to 269.

Question 7

The questionnaire with all questions should be added in the appendix

Response 7

I have attached the questionnaire and a consent form.

Question 8

The study limitations should be at the end of the discussion

Responces 8

I put the limitations at the end of the discussion.

We thank you for your help in improving the manuscript and if there are still aspects to be clarified, we will do so with pleasure.

Best regards,

Assoc. Professor, MD, PhD, Iulia Saveanu

Reviewer 3 Report

Line 22 – we should avoid to write about No of ethical acceptance in the abstract – it shoul be only mentioned in the main paper body in the Methods section

Line 25 – please write about only one gender e.x. female, because in this case there is more women

Line 25 – “from years II and III (38.2% and 61.8%, respectively)” – I don’t understand – what does this percent mean? You should clarify

Line 47 – “Transmission can occur”  - it’s not grammatically correct, maybe can be transmitted in this contect?

Line 69 – I agree that there is no vaccine available or post exposure drugs, but you should mention that DAA’s are very effcective, almost 100%

The abstract is too long – you should move some paragraphs to the discussion section or remove a few ones especially those with very basic medical knowledge

The questionnaire with all questions should be added in the appendix

The study limitations should be at the end of the discussion

Author Response

(The authors gave the same response as above.)

Round 2

Reviewer 1 Report

All my concerns have been addressed.

Author Response

Dear Reviewer,

Thank you for the evaluation and we wish you a rich scientific activity and excellent results!

I responded to the requests of the second evaluator, as follows.

Responses  1

I have specified bibliographic sources where necessary.

Responses  2

I reorganized the introduction, limited it, and supplemented the sources where necessary.

Responses 3

I have completed the study limitations

Responses 4

I have reformulated the conclusions.

Best regards,

Assoc. Prof. MD, PhD Saveanu Catalina Iulia

Reviewer 2 Report

The present study is a very interesting work, base on a questionnaire survey, regarding the blood-borne infection control . However, followings should be considered:

  1. There are paragraphs for which the bibliographic source is missing. Please check carefully that all the information provided and which is not a direct part of the study conducted by the authors has a bibliographic source. Example: „According to the estimates of the Centers for Disease Control and Prevention (CDC), 35 5.6 million people working in the healthcare...” – reference missing
  2. The introduction part is too long. I recommend reformulating the introduction part and reducing it to a maximum of one page, which summarizes the existing information in the literature that is really relevant for the present paper.
  3. The Study Limitations section is treated superficially.
  4. The part of the conclusions should be completely reformulated, it should state the most important outcome of your work. Attention, the Conclusion section, should not simply summarize the points already made in the main text, in this section the authors should provide an interpretation of the findings.

Author Response

Dear Reviewer,

Thank you for the evaluation and for these recommendations. With your help, the content of the article will be better structured.

Q1

There are paragraphs for which the bibliographic source is missing. Please check carefully that all the information provided and which is not a direct part of the study conducted by the authors has a bibliographic source. Example: "According to the estimates of the Centers for Disease Control and Prevention (CDC), 35 5.6 million people working in the healthcare..." - reference missing

Responses  1

I have specified bibliographic sources where necessary.

Q2

The introduction part is too long. I recommend reformulating the introduction part and reducing it to a maximum of one page, which summarizes the existing information in the literature that is really relevant for the present paper.

Responses  2

Thank you for your attention. I reorganized the introduction, limited it, and supplemented the sources where necessary.

Q3

The Study Limitations section is treated superficially.

Responses 3

I have completed the study limitations

 Q4

The part of the conclusions should be completely reformulated, it should state the most important outcome of your work. Attention, the Conclusion section, should not simply summarize the points already made in the main text, in this section the authors should provide an interpretation of the findings.

Responses 4

Thank you for this remark. I have reformulated the conclusions.

Best regards,

Assoc Prof. MD, PhD Saveanu Catalina Iulia
